# Investigating Photo-Degradation as a Potential Pheromone Production Pathway in Spotted Lanternfly, *Lycorma delicatula*

**DOI:** 10.3390/insects14060551

**Published:** 2023-06-13

**Authors:** Hajar Faal, Isaiah J. Canlas, Allard Cossé, Tappey H. Jones, Daniel Carrillo, Miriam F. Cooperband

**Affiliations:** 1Forest Pest Methods Laboratory, USDA—APHIS—PPQ S&T, 1398 W. Truck Rd., Buzzards Bay, MA 02542, USA; 2Tropical Research and Education Center, University of Florida, 18905 SW 280 St., Homestead, FL 33031, USA; 3Department of Chemistry, Virginia Military Institute, Lexington, VA 24450, USA

**Keywords:** *Lycorma delicatula*, photo-degradation, behavioral bioassays, pheromones, attractants

## Abstract

**Simple Summary:**

Evidence suggesting that the invasive species spotted lanternfly (SLF), *Lycorma delicatula*, may use pheromones has previously been shown. We hypothesized that sunlight might play a key role in SLF pheromone production based on observations that SLF adults spend time in the upper canopies of their host trees and their affinity for ultraviolet light. In this study, extracts from SLF nymphs and adults were either exposed to simulated sunlight (photo-degradation), or not exposed to light (crude), while volatiles were collected. Attraction of SLF nymphs and adults to volatiles from photo-degraded and crude samples, and their residues, was measured in dual-choice bioassays. SLF nymphs and adults responded differently to either residues or volatiles of both crude and photo-degraded body extracts. Photo-degradation did not increase the attraction of SLF individuals tested in two-choice bioassays, except for third instars. However, SLF males chose the residue from photo-degraded extracts over the residue from crude extracts of females. The results suggest that photo-degradation may play a role in sex-specific short-range communication, but photo-degradation did not appear to play a substantial role in long-range intraspecific chemical communication in SLF. This study produced additional evidence that SLF likely use pheromones in aggregation and mating.

**Abstract:**

Since its discovery in North America in 2014, the spotted lanternfly (SLF), *Lycorma delicatula*, has become an economic, ecological, and nuisance pest there. Developing early detection and monitoring tools is critical to their mitigation and control. Previous research found evidence that SLF may use pheromones to help locate each other for aggregation or mating. Pheromone production necessitates specific conditions by the insects, and these must be investigated and described. A chemical process called photo-degradation has been described as a final step in the production of pheromones in several diurnal insect species, in which cuticular hydrocarbons were broken down by sunlight into volatile pheromone components. In this study, photo-degradation was investigated as a possible pheromone production pathway for SLF. Extracts from SLF mixed-sex third and fourth nymphs and male or female adults were either exposed to simulated sunlight to produce a photo-degradative reaction (photo-degraded), or not exposed to light (crude), while volatiles were collected. Behavioral bioassays tested for attraction to volatiles from photo-degraded and crude samples and their residues. In third instars, only the volatile samples from photo-degraded mixed-sex extracts were attractive. Fourth instar males were attracted to both crude and photo-degraded residues, and volatiles of photo-degraded mixed-sex extracts. Fourth instar females were attracted to volatiles of crude and photo-degraded mixed-sex extracts, but not to residues. In adults, only males were attracted to body volatiles from crude and photo-degraded extracts of either sex. Examination of all volatile samples using gas chromatography coupled with mass spectrometry (GC-MS) revealed that most of the identified compounds in photo-degraded extracts were also present in crude extracts. However, the abundance of these compounds in photo-degraded samples were 10 to 250 times more than their abundance in the crude counterparts. Results from behavioral bioassays indicate that photo-degradation probably does not generate a long-range pheromone, but it may be involved in the production of a short-range sex-recognition pheromone in SLF. This study provides additional evidence of pheromonal activity in SLF.

## 1. Introduction

The spotted lanternfly, *Lycorma delicatula* (White) (Hemiptera: Fulgoridae) (hereafter SLF), is a polyphagous invasive pest in the United States [1], Korea [2], and Japan [3]. Since its first detection in the United States in 2014, infestations now occur in 14 states, including Pennsylvania, New Jersey, New York, Connecticut, Ohio, Delaware, Virginia, West Virginia, Maryland, Massachusetts, Rhode Island, Indiana, Michigan, and North Carolina [4]. This sap feeder has over 100 host plants and causes serious economic damage to vineyards, and their presence threatens other agricultural, timber, and nursery industries [5,6,7,8]. Their broad host range, ease of spread via hitchhiking over long distances [9,10], and a poor understanding of their biology and ecology highlight the need for more research, which in turn may facilitate the development of tools for early detection, monitoring, and control.

Fortunately, our understanding of SLF biology and ecology is rapidly improving [8,11,12,13,14,15,16,17,18,19,20], and shedding light on their behavior and chemical ecology [21,22,23,24]. Until recently, attraction of SLF to conspecific-derived chemical compounds such as pheromones was unknown [21], but our recent work has provided the first evidence that such compounds may exist [18,19]. Adult male SLF are attracted to male-produced honeydew volatiles [19] and male- and female-produced body volatiles [18]. The conditions required to stimulate the production of any such pheromones remain unknown.

Recent studies on SLF have hinted that sunlight may play a role in SLF biology and behavior. In one study (Cooperband, unpublished), SLF adults were found to reside at different heights in the canopy at different developmental times, suggesting that during mating both sexes preferred the upper canopy [25]. This behavior could occur if SLF were attracted to sunlight during mating time. A study on the diurnal activity of SLF found that the majority of SLF reproductive activities (mating pairs, courtships, and ovipositing) have been observed in early afternoon between 12:00–15:00 [12,21] when the intensity of sunlight and ultraviolet radiations is maximum [26]. In another study, SLF were found to orient towards ultraviolet light (395–410 nm wavelengths) in laboratory bioassays [27]. Ultraviolet light can breakdown unsaturated cuticular hydrocarbons of insects and result in volatile organic compounds [28]. Cuticular hydrocarbons are aliphatic material that cover the outer layer of insect body surface [29] and play an important role in species and sex recognition [30]. Interestingly, it has also been reported that aliphatic (long-chain) cuticular hydrocarbons of male and female SLF chemically differ in their presence of 21- to 33-carbon molecules with methyl groups [31], and related cuticular hydrocarbons have been reported in other insects as precursors that convert to pheromones in the presence of sunlight [32,33,34].

Sunlight has been found to play an essential role in pheromone production in some species of insects through a photo-degradative process called photo-oxidation [32,33,35]. Photo-degradation is a physical and chemical alteration of photodegradable chemicals caused by the absorption of photons, particularly infrared radiation and ultraviolet light found in sunlight [36]. In this process, long-chain chemicals with double bonds break down into short-chain chemicals. Oxygen may (photo-oxidative) or may not (chain-breaking) be involved in photo-degradation reactions [36]. The chemical mechanisms involved in oxidation is reviewed by Frankel [37]. Several sex pheromones have been identified as products of photo-oxidation in several insect taxa [34], including Hemiptera [38], Hymenoptera [35,39], Coleoptera [33], and Diptera [40]. The primary objective of this study was to investigate whether a photo-degradation pathway resulted in the production of SLF pheromones. This was accomplished by comparing SLF attraction to volatiles from SLF body extracts, after treatment with or without simulated sunlight. In this study, we targeted third instars of unknown sex, fourth instars of both sexes, and adults of both sexes.

Here we hypothesized that the waxy cuticle of SLF could be targeted by photo-degradative reactions, and questioned whether photo-degradation could (1) increase SLF attraction by (2) increasing the abundance of behaviorally active compounds in samples, also facilitating their subsequent analysis.

In the current study, we (1) tested volatile collections of crude (not exposed to light) and photo-degraded extracts, and their residues, for attraction in a dual-choice olfactometer; (2) screened volatile collections of crude and photo-degraded extracts for antennally active compounds using GC-EAD; (3) used GC-MS to identify the antennally active components of samples; (4) confirmed tentative GC-MS identifications using authentic standards; (5) described the chemical profiles of the antennally active volatiles collected from photo-degraded and non-photo-degraded (crude) SLF body extracts; and (6) evaluated individual synthetic standards of antennally active compounds for attraction in a dual-choice olfactometer.

## 2. Materials and Methods

### 2.1. Insect Collection and Maintenance

SLF third instars, fourth instars, and adults were field-collected from pesticide-free trees of *Ailanthus altissima* (Mill.) Swingle (Sapindales: Simaroubaceae) in Lehigh and Monroe Counties in Pennsylvania, USA, and Warren County in New Jersey, USA. Third and fourth instars were collected from the middle of June for three weeks and the beginning of July 2021 for two weeks, respectively. Adults were collected from the end of July to mid-October in 2020 and 2021. Field-caught insects were collected each Monday and shipped the same day to arrive Tuesday morning at the insect containment facility located at the Forest Pest Methods Laboratory (FPML) in Buzzards Bay, Massachusetts, USA, adhering to permit conditions from the Pennsylvania Department of Agriculture (PDA) (PP3-0123-2015) and the U.S. Department of Agriculture (USDA) (P526P-15-00152 and P526P-20-03198).

Upon arrival in the insect containment facility, SLF were transferred to rearing cages housed at 23.3 °C in a greenhouse room with natural daylight conditions supplemented with grow lights (TSL 2000 LED full spectrum, 2000 W, 25 × 100 cm coverage, Mars Hydro, Commerce, CA, US) set for 16:8 h L:D. Nymphal stages were housed in cages (30 × 30 × 30 cm, Bugdorm, Megaview Science Co., Ltd., Taichung City, Taiwan) containing freshly cut *A. altissima* branches maintained in hydroponic solution (Maxigrow, GenyHydro Inc., Sebastopol, CA, prepared according to label). SLF adults were kept in cages (47.5 × 47.5 × 93 cm, Bugdorm, Megaview Science Co., Ltd., Taichung City, Taiwan) containing whole potted plants of *A. altissima* (3 months old, 40 cm tall, 4 L pots). Up to 30 fourth instars and up to 20 adults were kept per cage. Adult males and females were maintained in separate cages. Potted host plants were replaced every two weeks, or earlier if plants showed signs of stress. These live SLF were used in laboratory behavioral bioassays and GC-EAD that were typically conducted Tuesday through Friday.

In addition to collecting live insects, separate weekly batches of SLF were flash-frozen as they were field-collected in order to preserve them for laboratory extraction. As such, those SLF were collected into pre-baked oven bags (Turkey size, Reynolds Consumer products, Field court, Lake Forest, IL, USA) lining plastic cups (946 mL) containing dry ice, which froze them instantly. Oven bags had been pre-baked at 150 °C for 4 h to remove volatile contaminants such as caprolactam [41]. Male and female nymphs were combined in frozen collections, but adults were frozen in separate containers based on their sex. They were shipped every Monday in a cooler with dry ice, arriving Tuesday morning at the FPML where they were immediately used to make insect extracts.

### 2.2. Whole Body Extract

Upon arrival at FPML, the flash-frozen SLF were placed into a 500 mL glass beaker, covered with dichloromethane (DCM) (>99.9% purity, Fisher Scientific, Fair Lawn, NJ, USA), then held at room temperature for 10 min. Then these whole-body extracts were decanted through glass wool, to exclude solids, into 20 mL scintillation vials. This procedure was used for mixed-sex third or fourth instars, and male and female adults (during the three adult physiological phases of “Early” (adults before mating), “Mid” (mating has begun), and “Late” (oviposition has begun). Any broken adults were quickly removed prior to extraction.

### 2.3. Photo-Degradation Process

Photo-degradation was conducted using SLF extracts from mixed-sex third instars, mixed-sex fourth instars, adult males, or adult females exposed to light from a solar simulator (LS series light source, Abet Technologies, Milford, CT, USA), as described below (Appendix A). For each stage and sex, whole-body extracts were divided into two equal portions, with half slated for photo-degradation and the other half to remain as crude extract (control). Each portion was then evaporated just to the point of dryness, as described in Bartelt et al. [32], in a quartz cuvette (1-Q-10-GL14-S, spectrophotometer wavelength range within 170 to 2700 nm, Starna Cells, Inc., Atascadero, CA, USA), leaving a waxy film on the interior cuvette walls. The cuvette vial was placed 10 cm from the light source and exposed to the solar simulator under ambient conditions at 25.5 °C overnight (19 h), during which time the headspace body volatiles (photo-degraded) were collected on HayeSep-Q traps (80–100 mesh, Hayes Separation Inc., Bandera, TX, USA) with filtered air flow (25 mL/min). Volatile collection traps consisted of 100 mg of HayeSep-Q packed into glass Pasteur pipettes, secured with glass wool, and pre-cleaned with DCM then baked.

Crude extracts were prepared and handled in the same manner as photo-degraded samples but the cuvette vials containing crude extracts were wrapped with a piece of aluminium foil and placed away from the solar simulator light. Thus, the crude headspace body volatiles were collected without light exposure. For both photo-degraded and crude samples, the residues coating the cuvettes were collected by rinsing with 1 mL DCM, and the headspace volatiles were collected by eluting the HayeSep-Q traps with 1 mL DCM containing 1 ng/µL of tricosane (Sigma-Aldrich) as an internal standard.

### 2.4. Lure Preparation

Under a gentle stream of nitrogen, the headspace volatiles and residues of extracts in DCM were concentrated to the desired number of insect equivalents (given below), which was calculated by dividing the total number of insects used for each extract by the volume of total extract. Behavioral bioassays testing live third or fourth instar SLF measured their responses to a choice between the residue and control (100 μL of clean DCM) or headspace volatiles and control. For adults, choices of both sexes were measured in response to the headspace volatiles from each sex compared to the control, which essentially doubled the number of assays and resulted in time constraints. Therefore, most adult bioassays focused only on responses to the headspace volatiles, not the less volatile residues, since the ultimate goal was to identify long-range volatile attractants. All lures consisted of 21.3 ± 0.9 insect equivalents in 100 μL of DCM in an open 0.5 mL microcentrifuge tube (Globe Scientific INC., Mahwah, NJ, USA) or 100 μL DCM controls, with the exception of a set of preliminary behavioral bioassays testing a head-to-head choice between photo-degraded and crude extract residues of Early adults, which used lures containing 8.6 ± 0.4 insect equivalents in 100 μL DCM lures. For bioassays evaluating single synthetic compounds, lures consisted of open microtubes containing 1 mg of neat material and were compared to empty control tubes. All lures were prepared immediately before use in the behavioral bioassays.

### 2.5. Gas Chromatography—Mass Spectrometry (GC-MS)

All mass spectrometry analyses were conducted on headspace volatiles collected from either crude or photo-degraded extracts, concentrated to 1 insect equivalent per μL. These were analyzed with an Agilent 7890B GC coupled to an Agilent 5977A MS (EI mode, 70 eV with a scanning range of 40.0–450.0 *m*/*z*), equipped with a DB-5MS capillary column (Agilent, 30 m×0.25 mm i.d., 0.25 µm film thickness) and helium carrier gas at a constant flow rate of 1 mL/min. The 7693 autosampler (Agilent Technologies) was used for sample injections (1 μL) in splitless mode with a 250 °C injection port and an oven temperature of 40 °C for 2 min, increasing at 5 °C/min to 300 °C. Tentative identifications of compounds from the library database match (Enhanced ChemStation, MSD Chemstation, Data Analysis software vF.01.00.1903, and NIST, v11, Agilent Technologies, Santa Clara, CA, USA) were then verified according to their mass spectrum, Kovat’s indices, and co-chromatography with synthetic standard compounds purchased from Sigma-Aldrich (St. Louis, MO, USA), except (*E*)-non-2-enal (Fisher Scientific, Hampton, NH, USA), (*Z*)-tetradec-4-ene (synthesized by THJ), and octane-2,3-dione, which was synthesized (by THJ) as described by Pfeifer and Kroh [42] (Appendix A). The total abundance of each tentatively identified compound in adult males and females were estimated by relating ion abundance peak areas to that of the internal standard.

### 2.6. Gas Chromatography—Electroantennographic Detection (GC-EAD)

All volatile samples were screened on a GC-EAD system for the presence of bioactive compounds using the antennae of third instars, fourth instars, and adults. GC-EAD analyses were carried out on a 6890 GC (Agilent) fitted with a flame ionization detector (FID, 250 °C) and HP-5MS column (30 m × 0.320 mm I.D. × 0.25 μm film, Agilent Technologies, Inc., Santa Clara, CA, USA). The carrier gas was helium (1 mL/min), the injection was splitless, and the oven temperature program was as described for the GC-MS. The column effluent was split 1:1 with a glass Y-connector (Agilent Technologies) between the FID and the EAD [43]. The EAD effluent passed through a heated transfer line (250 °C, Syntech Temperature Controller, Kirchzarten, Germany) into a glass, L-shaped, odor delivery tube (11 mm i.d.) through which air passed (300 mL/min) and was delivered to the insect antenna. All volatile samples were injected at a concentration of 1 insect equivalent in 1 μL, half of which was delivered to the insect antennae. For each sample, five replicates were performed using the antennae of each stage and sex.

For antennal preparation, the SLF head was mounted onto a ground electrode, formed from a custom-pulled glass capillary containing Ringer’s solution [23]. The tip of the arista was clipped, and the remaining arista was inserted into a pulled-glass capillary recording electrode filled with Ringer’s solution. Antennal signals were amplified using a Dam 50 differential amplifier (World Precision Instruments, Sarasota, FL, USA), passed through Hum Bug 50/60 Hz noise elimination (Quest Scientific, North Vancouver, BC, Canada), and integrated with a two-channel signal acquisition interface (IDAC-2, Syntech, Hilversum, The Netherlands). Signals were collected and analyzed using GcEad/2014 software (Syntech, Version 1.2.5, Kirchzarten, Germany).

### 2.7. Behavioral Bioassays

All behavioral bioassays were performed using two different sizes of custom Teflon Y-plate olfactometers, as described in [22] in an environmental chamber at the FPML Insect Containment Facility, with an average temperature of 22 ± 0.3 °C (Appendix A). The small Y-plates were used for third instars (16.5 cm long × 12.7 cm wide × 1.3 cm tall, with 1.9 cm wide channel) [44], and the large Y-plates were used for fourth instars and adults (28.6 cm long × 21.6 cm wide × 3.8 cm tall, with 5.1 cm wide channel) [22]. On both sides of the olfactometer, the charcoal-filtered humidified air passed through a 50 mL glass flask prior to entering the arm of the olfactometer. A lure containing the stimulus or the control was placed in the flasks. Air velocity exiting the bottom of the Y-plate was measured using a hot wire anemometer (Testo 405 digital mini anemometer with hot-wire probe, Testo North America, West Chester, PA, USA) during each session and was 24 cm/s in both Y-plates.

The bioassay methodology, as described below, was identical to previously described methods [18,19,22,23]. Prior to each bioassay session, five SLF were tested in a clean Y-plate without any lures to ensure that there was no directional bias in the system. Prepared lures (extracts or control) were placed in the airflow upwind of the two arms of the Y-plate bioassay at the start of each bioassay session. Then, a single SLF was released into the bottom of the Y and was allowed 3 min to make a choice. A choice was recorded if the insect walked past the split and more than halfway to the end of one of the upwind arms, at which point the insect was removed. If no choice was made in 3 min, the insect was removed and was recorded as no choice. This procedure was replicated with a new insect up to 20 times per session. A maximum of 10 males and 10 females, alternating, were tested in each session. To achieve higher numbers of replicates, additional sessions were conducted with fresh Y-plates and the directions of the stimuli were reversed. Prior to each session, Y-plates were washed with warm water and odorless detergent (Alconox Inc., White Plains, NY, USA), rinsed with water then 95% ethanol, allowed to dry overnight in a hood, and disposable acetate Y-plate ceiling and floor were replaced [22,23].

All extracts obtained from third or fourth instars were mixed-sex extracts. The behavioral bioassays tested third instars of unknown sex, and fourth instar males and females in four series: (1) residue of crude extracts vs. DCM control, (2) residue of photo-degraded extracts vs. DCM control, (3) headspace volatiles of crude extracts vs. DCM control, and (4) headspace volatiles of photo-degraded extracts vs. DCM control.

Choices of Early males and females in response to Early male and Early female body volatiles were recorded in three series of experiments: (1) residue of photo-degraded extracts vs. residue of crude extracts, (2) headspace volatiles of crude extracts vs. DCM control, and (3) headspace volatiles of photo-degraded extracts vs. DCM control.

Finally, a subset of EAD-active, individual synthetic compounds was evaluated in behavioral bioassays for responses by female and male fourth instars and adults, comparing 1 mg of neat material to empty control tubes.

Statistical analysis was conducted using a Chi Square test and the null hypothesis that both lures would be chosen at equal frequencies. A test statistic of G greater than 3.84 resulted in a rejection of the null hypothesis and a significant difference between the two choices [45].

## 3. Results

### 3.1. GC-MS Analysis

Analysis of headspace volatile samples by GC-MS revealed that the photo-degradation process resulted in an increase in both the amount and number of eluting components compared to crude extracts (Figure 1 and Figure 2). There were numerous unknown compounds in the extracts, but a total of 48 compounds were tentatively identified by the NIST Library database match, and comparison of mass spectra and of Kovat’s indices to those in the literature (Figure 1). From those, 33 compounds were identified using synthetic standards: octane, undecane, tridecane, tetradecane, undec-1-ene, dodec-1-ene, tridec-1-ene, pentadec-1-ene, hexanal, heptanal, octanal, nonanal, decanal, pentan-1-ol, hexan-1-ol, heptan-1-ol, octan-1-ol, nonan-1-ol, oct-1-en-3-ol, 2-ethylhexan-1-ol, heptan-2-one, octan-2-one, nona-2-one, octane-2,3-dione, methyl 2-hydroxybenzoate, hexanoic acid, heptanoic acid, octanoic acid, nonanoic acid, (*Z*)-tetradec-4-ene, (*Z*)-non-6-enal, (*E*)-non-2-enal, and (*E*)-dec-2-enal. Although the positions and *E/Z* configurations of double bonds were not chemically verified for (*Z*)-tetradec-4-ene, (*Z*)-non-6-enal, (*E*)-non-2-enal, and (*E*)-dec-2-enal, their retention times and mass spectra matched with those of authentic compounds. In addition, the authentic standards of these were used in bioassays.

Most tentatively identified and unknown compounds were found in photo-degraded extracts across different stages (Figure 2 and Figure 3). The volatile profiles of photo-degraded extracts of nymphal stages (third and fourth instars), females (Early, Mid, Late), and males (Early, Mid, Late) had 47, 44, 45, 45, 46, 48, 46, and 44 of 51 compounds, respectively (Figure 3). Crude extracts contained fewer of these compounds and in much lower abundance relative to those in corresponding photo-degraded extracts (Figure 2 and Figure 3). The volatile profiles in crude extracts of nymphal stages (third and fourth instars), females (Early, Mid, Late), and males (Early, Mid, Late) had 32, 19, 15, 7, 7 25, 21, and 7 of these 51 compounds, respectively (Figure 3).

Several were found only in photo-degraded extracts: pentan-1-ol, octane, (3*E*)-octa-1,3-diene, (*E*)-hex-2-enal, hexan-1-ol, heptan-1-ol, octane-2,3-dione, (*E*)-oct-3-en-2-one, (*E*)-oct-2-enal, (*Z*)-oct-2-en-1-ol, nonan-1-ol, and undecan-1-ol (Figure 2 and Figure 3). Several compounds were present in certain adult phases of both crude and photo-degraded extracts. For instance, volatile profiles from Early males and Early females contained (*E*)-non-4-enal in both crude and photo-degraded extracts, and it was also found in crude extracts of Late males. (*Z*)-non-6-enal appeared only in volatile collections from crude and photo-degraded extracts of Late females and photo-degraded extracts of Late males.

### 3.2. GC-EAD Analysis

In GC-EAD analyses, 35 components in the headspace volatiles of photo-degraded extracts evoked antennal responses in third instars, fourth instars, and adults (Appendix A). In addition, the EAD-activity of five unknown compounds remained inconclusive (Figure 1, peaks number 13–17). Most of these compounds occurred in both PD and crude extracts, the quantities in crude extracts were much lower, and only those which appeared as major components in crude extracts generated antennal responses there. The abundance of these compounds in photo-degraded samples was 10 to 250 times more than their abundance in the crude counterparts (Figure 2).

From all EAD-active compounds, twenty-eight of those were confirmed with synthetic standards, including hexanal, heptanal, octanal, nonanal, decanal, (*E*)-non-2-enal, (*Z*)-non-6-enal, (*E*)-dec-2-enal, 2-ethylhexan-1-ol, hexan-1-ol, heptan-1-ol, octan-1-ol, nonan-1-ol, oct-1-en-3-ol, heptan-2-one, octan-2-one, nona-2-one, undec-1-ene, dodec-1-ene, tridec-1-ene, (*Z*)-tetradec-4-ene, pentadec-1-ene, undecane, tridecane, hexanoic acid, heptanoic acid, methyl 2-hydroxybenzoate, and octane-2,3-dione; four were tentatively identified as oct-1-en-3-one, (*E*)-oct-3-en-2-one, (*Z*)-oct-2-en-1-ol, and undecane-1-ol; three compounds remained unknown (Figure 1, peaks number 9, 29, and 40). The synthetic standards of several compounds found in the headspace body volatiles of SLF also produced antennal responses in both sexes: pentan-1-ol, octane, octanoic acid, and tetradecane.

### 3.3. Behavioral Bioassays

Third instar SLF of unknown sex were evaluated for their preferences in response to a choice between the headspace volatiles or their residues of mixed-sex third instar extracts, that were either photo-degraded or crude, compared to a solvent control. For third instars, only headspace volatiles of photo-degraded extracts were significantly attractive in the dual-choice bioassay, with a relatively low response rate of 47.5% (Figure 4). The highest response rate was 60% to residue of PD extract, but without a significant preference over the control.

Bioassays involving fourth instars similarly used mixed-sex extracts, but the live insects tested in the bioassay were sexed before use. Fourth instar SLF males were significantly attracted to all stimuli tested, except to headspace volatiles of crude extracts. Fourth instar females were significantly attracted to the volatiles from both crude and photo-degraded extracts, but not to their residues (Figure 5). Response rates ranged from 57.5% to 75%, with the highest response rate associated with no preference by females for residue of PD or control.

The first series testing Early adult behavior evaluated male and female responses when presented with a head-to-head choice between photo-degraded and crude extract residues in a Y-plate bioassay (Figure 6). In that series, the photo-degraded residues from Early females were significantly more attractive to Early males than crude residues from Early females (Figure 6). No preferences were observed between crude and photo-degraded extract residues of either sex by females, or of males by males (Figure 6).

The remaining two series tested Early adult responses when offered a choice between headspace volatiles and controls. Early adult males were significantly attracted to both crude (series 2) and photo-degraded (series 3) volatiles from both sexes, but none of these volatile samples were attractive to Early females (Figure 6).

Although GC-EAD reveals which compounds can be detected by the insect antenna, it cannot inform us about the behavioral function of a compound. It was not possible to test each of the 28 antennally active compounds for behavior individually and to each stage and sex, given time, space, insect availability, and personnel constraints. We attempted to test compounds for behavior with the stage from which it was found, but this was not always possible due to time and logistical constraints. In some cases, compounds found in adults were tested on nymphs the subsequent season, at the time not yet having the full picture of which stages produced which compounds since the final analysis was not yet complete. Therefore, a few compounds were selected to test individually for behavioral function in available stages.

In these bioassays, fourth instar males and females and adult males and females in different physiological states were offered a choice between a synthetic compound and a blank control. In fourth instars and adults, the preferences of SLF males were found to be different from those of females in response to different compounds. For instance, in fourth instars, (*Z*)-non-6-enal was attractive only to females, whereas (*Z*)-tetradec-4-ene and tridec-1-ene were attractive only to males. In adults, octane-2,3-dione and (*E*)-non-2-enal were attractive to Early and Late males, respectively, but not females, and (*E*)-dec-2-enal was attractive only to Early females but not males. Several compounds produced behavioral responses from both adult sexes. For example, undecane was attractive to both Early males and females. Interestingly, nona-2-one, which previously was found to attract adult Early females but not Early males [19], attracted both adult Mid females and males (Figure 7).

## 4. Discussion

In this set of experiments, we explored whether photo-degradation could generate volatile and semi-volatile pheromone components in SLF. Generally, while volatile pheromones can be used for long-range insect communication and orientation, semi-volatile or non-volatile pheromones can be used in close range communication and may be involved in such activities as contact chemoreception, close-range orientation, marking, trails, recognition, and courtship [46]. The volatility of the compound will relate to the maximum distance of the communication [47]. In the current study, headspace samples contained volatile components, while residues included semi-volatile and non-volatile components of body extracts. We have shown that the preferences of SLF differed in behavioral bioassays depending on insect stage, sex, volatile source, and the exposure of extracts to the solar simulator.

In third instars, only the volatile samples from photo-degraded mixed-sex extracts were attractive, suggesting the possible presence of a long-range attractant used in aggregation that is upregulated by sunlight. Fourth instar females were attracted to volatiles of crude and photo-degraded mixed-sex extracts but not to residues, suggesting the possible presence of a long-range signal used by fourth instar females for aggregation. Fourth instar males were attracted to both crude and photo-degraded residues, suggesting they may utilize a short-range recognition signal. In addition, fourth instar males were attracted to volatiles of photo-degraded mixed-sex extracts, suggesting the presence of a long-range signal used for aggregation that is upregulated by sunlight. A study by Cooperband et al. [48] demonstrated that marked SLF nymphs of all stages oriented to aggregations of other SLF nymphs in the field, and our results here offer a possible aggregation pheromone mechanism for that behavior.

In adults, only males were attracted to body volatiles from crude and photo-degraded extracts of either sex, suggesting adult males, but not females, may use a volatile signal produced by both sexes (regardless of sunlight) to locate and orient to aggregations. In addition, male-to-female attraction was observed in response to residues, and photo-degradation appeared to enhance that attraction in residues but not in volatiles, suggesting the presence of a close-range sex-recognition pheromone that is upregulated by sunlight, produced by females and used by males. Long-range attractants may help SLF males locate patches of conspecifics for aggregation. Subsequently, since SLF then spend much of their time walking on trees among aggregations, it makes sense that some of their conspecific chemical communication might involve semi-volatile pheromones that would potentially be used in species recognition, sex-recognition, courtship, or close-range orientation, particularly by mate-seeking males [46]. On the other hand, adult females did not appear to respond to volatiles originating from either adult males or females. Only fourth instar females showed attraction to mixed-sex conspecific body extract volatiles.

Before mating (Early), males that were offered a choice between residues that were either photo-degraded or crude significantly chose extracts from Early females (not males) that were photo-degraded over those that were crude. Because residues consist of mostly the non-volatile and semi-volatile portion of the extract, this could be evidence of a possible close-range or contact sex-recognition pheromone, which may undergo its final production step during photo-degradation, although alternative explanations may exist as well. For instance, the greater attraction to photo-degraded extract residue over crude extract residue could be due to the presence of a greater amount or number of pheromones as a result of photo-degradation. Derstine et al. [23] showed that more SLF were attracted to a single compound at a larger dose than a smaller dose, and SLF preferred multi-component blends over an individual compound. The observed male preference for the female extract residue which was photo-degraded might be explained by either the production of a new compound, or by an overall increase in pheromones or abundance. Unfortunately, we did not analyze the makeup of residues due to their heavy waxy content [18,31,49,50], and it was beyond the scope of this project which targeted long-range volatile attractants, not the less-volatile compounds found in residues.

Compared to solvent controls, the volatiles of photo-degraded extracts of either sex attracted similar levels of males as their corresponding crude extracts, suggesting sunlight is likely not needed to generate long-range aggregation pheromones. Examination of all volatile samples using GC-MS revealed that most of the identified compounds in photo-degraded extracts were also present in crude extracts but in lower quantities. This occurred across different stages (third and fourth instars, Early, Mid, and Late adults) and, with three exceptions, none of the compounds were sex- or stage-specific. The exceptions included undecane, found in third and fourth instars, and (*E*)-non-4-enal and (*Z*)-non-6-enal, found in male and female adults (Early and Late adults, respectively). However, with roughly one insect equivalent injected into the GC-MS, the small amounts of some compounds approached the lower detection range of our instruments. It is possible that their abundance in some samples could have been below our detection level. The electroantennogram results also showed that both male and female antennae detected the same compounds without showing any compound unique to either sex. Despite the similarities in chemical composition of male and female body extracts and antennal activities of both sexes, only adult males, but not adult females, were attracted to headspace volatiles from crude and photo-degraded extracts of males or females.

We hypothesized that sunlight might play a key role in SLF pheromone production based on observations that SLF adults spend time in the upper canopies of their host trees during mating [25], their reproductive activities in early afternoon [12], their affinity for ultraviolet light [27], and reported differences between the composition of male and female long-chain cuticular hydrocarbons [31]. Under sunlight, unsaturated lipids degrade to short-chain volatile hydrocarbons [32,37], which could act as pheromones for species and sex recognition [30]. The results from our behavioral bioassays with volatiles and residues from whole body extracts suggest that photo-degradation may play a role in sex-specific short-range communication, but it is unlikely to play a substantial role in long-range intraspecific chemical communication in SLF. Nonetheless, we learned from GC-MS analyses that photo-degradation enhanced the amount of volatile compounds existing in crude extracts to detectable levels for SLF EAD. This process might be useful to facilitate identification of unknown chemicals in crude extracts. We did not scrutinize the possibility of thermal degradation during sample processing. Small molecules are subject to a wide range of temperatures during storage in freezer (−20 °C) or analyzing on GC-MS (300 °C), which could lead to the formation of degradation products [51].

We explored the behavioral function of several individual compounds in dual-choice assays, and found that SLF males and females responded differently to the same chemicals. Of the three compounds evaluated for responses by fourth instars, female fourth instars were attracted only to (*Z*)-non-6-enal, whereas (*Z*)-tetradec-4-ene and tridec-1-ene were attractive only to male fourth instars. Of the five compounds tested in behavioral bioassays for adult responses, octane-2,3-dione and (*E*)-non-2-enal, were attractive only to males, and (*E*)-dec-2-enal was attractive only to females. Early adult males and females were both attracted to undecane. Interestingly, although nona-2-one was only attractive to Early females, but not to Early males, both Mid males and females were attracted to it. Three components, heptan-2-one, octan-2-one, and nonan-1-ol, were previously found in SLF honeydew as well [19]. In that study, behavioral responses between adult males and females differed in that heptan-2-one was attractive only to males, octan-2-one was attractive only to females, and nonan-1-ol, repelled only adult females [19]. Several EAD-active components were also found in volatiles of UV-exposed body extracts from American cockroach, including nona-2-one, pentan-1-ol, octan-1-ol, nonan-1-ol, and hexanoic acid. A mixture of these compounds generated either no responses or aggregation responses at low or high concentrations, while a dispersal behavior was observed when the mixture was combined with a series of fatty acids [28]. Our dual-choice assays with individual compounds indicate that same compound could generate different signals to SLF males and females. However, we did not pursue describing the function and importance of each identified compound across different stages due to the number of antennally active compounds and time and resource constraints. GC-EAD is a system to screen the antennal activity of compounds, and it does not provide the behavioral function of those compounds for the insects tested [52,53,54]. It is notable that the antennae of both sexes detected the same set of compounds while male and female behavioral responses to those compounds differed. The ability of male and female antennae to detect the same range of compounds is a general phenomenon in phytophagous insects [55,56]. However, this phenomenon could be explained by (1) differences in the meaning of each compound to each sex and (2) differences in male and female SLF antennal sensitivity, which are currently being investigated.

## 5. Conclusions

SLF were attracted to conspecific volatiles in behavioral bioassays. Photo-degradation enhanced attraction of male SLF to female extract residues which contained the non-volatile or semi-volatile components of extracts. Although photo-degradation of SLF extracts increased the amount of antennally active SLF-derived pheromones, it did not result in a substantial increase in SLF attraction to the volatile portion of the extracts, suggesting that sunshine is probably not involved in a long-range pheromone production pathway for SLF, but may be involved in a short-range sex-recognition pheromone. Future research targeting close-range sex-recognition pheromones should seek to analyze photo-degraded residues to identify the compound(s) responsible for the observed attraction. This study provides additional evidence of conspecific chemical communication in SLF and describes SLF chemical profiles that may function in combination with other modes of communication.

## Figures and Tables

**Figure 1 insects-14-00551-f001:**
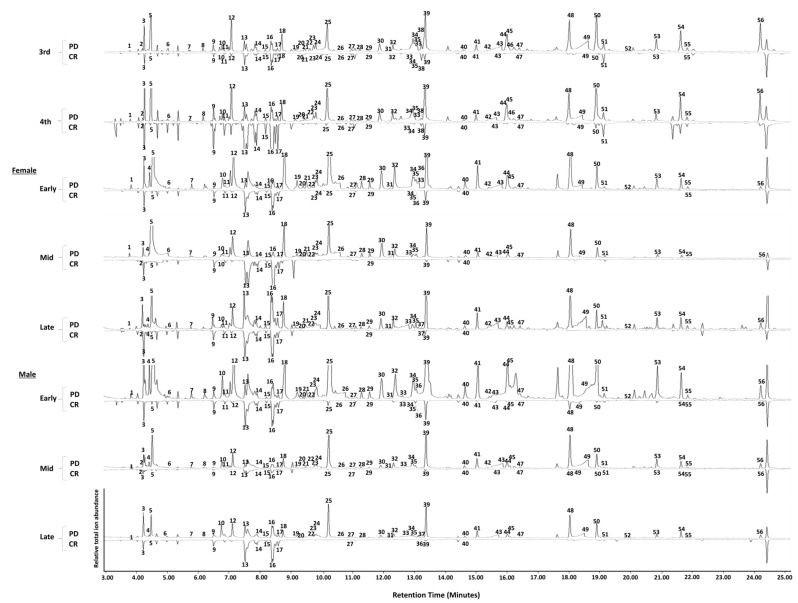
Relative total ion abundance of the headspace volatiles of either photo-degraded (PD) or crude (CR) extracts from different stages and sexes of spotted lanternflies, *Lycorma delicatula*, using gas chromatography–mass spectrometry (GC-MS). Peaks labeled with numbers include 48 tentatively identified compounds, as well as 8 unknown antennally active compounds. Compounds are numbered as follows: 1. pentan-1-ol; 2. hexan-2-one; 3. oct-1-ene; 4. octane; 5. hexanal; 6. (3*E*)-octa-1,3-diene; 7. (*E*)-hex-2-enal; 8. hexan-1-ol; 9. unknown; 10. heptan-2-one; 11. non-1-ene; 12. heptanal; 13–17. unknown; 18. (*Z*)-hept-2-enal; 19. heptan-1-ol; 20. oct-1-en-3-one; 21. oct-1-en-3-ol; 22. octane-2,3-dione; 23. octan-2-one; 24. dec-1-ene; 25. octanal; 26. hexanoic acid; 27. 2-ethylhexan-1-ol; 28. (*E*)-oct-3-en-2-one; 29. unknown; 30. (*E*)-oct-2-enal; 31. (*Z*)-oct-2-en-1-ol; 32. octan-1-ol; 33. heptanoic acid; 34. nona-2-one; 35. undec-1-ene; 36. (*E*)-non-4-enal; 37. (*Z*)-non-6-enal; 38. undecane; 39. nonanal; 40. unknown aldehyde; 41. (*E*)-non-2-enal; 42. nonan-1-ol; 43. octanoic acid; 44. decan-2-one; 45. dodec-1-ene; 46. methyl 2-hydroxybenzoate; 47. decanal; 48. (*E*)-dec-2-enal; 49. nonanoic acid; 50. tridec-1-ene; 51. tridecane; 52. undecan-1-ol; 53. undec-2-enal; 54. (*Z*)-tetradec-4-ene; 55. tetradecane; 56. pentadec-1-ene.

**Figure 2 insects-14-00551-f002:**
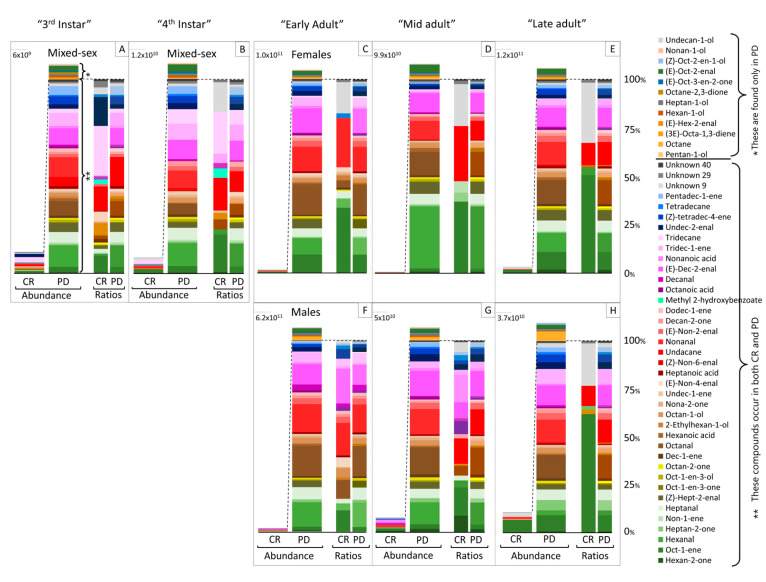
Relative abundance and ratios of tentatively identified compounds in the headspace volatiles of either crude (CR) or photo-degraded (PD) extracts of spotted lanternflies, *Lycorma delicatula*, from third (**A**) and fourth instars (**B**), adult females during Early (**C**), Mid (**D**), Late (**E**), and adult males during Early (**F**), Mid (**G**), and Late (**H**) using gas chromatography–mass spectrometry (GC-MS). The 12 compounds marked by a single asterisk (*) were only found in PD, whereas the remaining compounds (**) were found in both CR and PD.

**Figure 3 insects-14-00551-f003:**
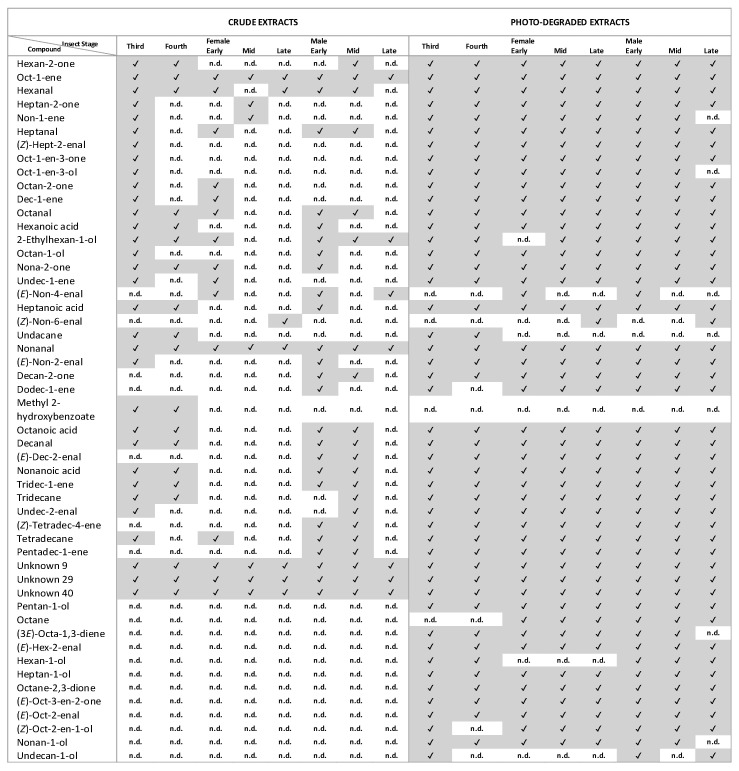
Absence or presence of tentatively identified and unknown compounds in crude and photo-degraded extracts across different stages. Absence and presence of compounds are indicated by n.d. (not detected) and check marks, respectively.

**Figure 4 insects-14-00551-f004:**
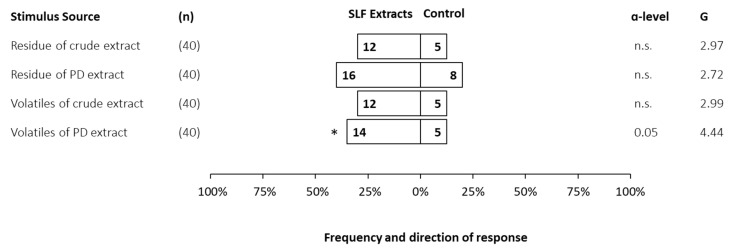
Frequency and direction of responses made by third instar spotted lanternflies, *Lycorma delicatula*, when offered a choice between volatiles of either crude or photo-degraded (PD) extracts (21.3 ± 0.9 insect equivalents) and solvent control (dichloromethane). The total number of insects tested (n), the number of responsive insects to the respective choice (numbers in bars), α-level below which *p*-values fell, and G-value are included for each test. Statistically significant choices are marked with an asterisk (*), and non-significant choices are represented as n.s.

**Figure 5 insects-14-00551-f005:**
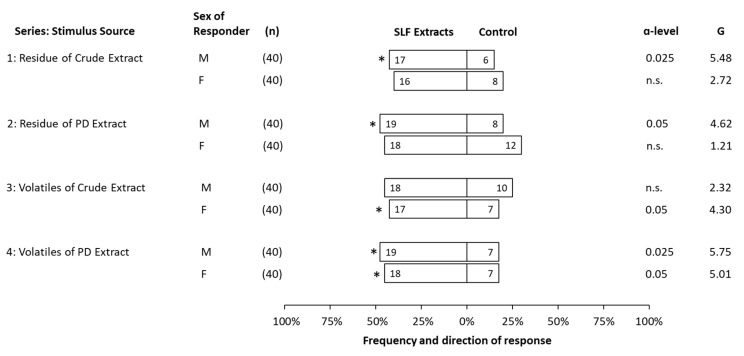
Behavioral responses in a dual-choice olfactometer made by male (M) or female (F) fourth instar spotted lanternflies, *Lycorma delicatula*, when offered a choice between either residue or volatiles of either crude or photo-degraded (PD) extracts (21.3 ± 0.9 insect equivalents) compared to a solvent control (dichloromethane). The total number of insects tested (n), the number of responsive insects to the respective choice (numbers in bars), α-level below which *p*-values fell, and G-value are displayed for each test. Statistically significant choices are marked with an asterisk (*), and non-significant choices are represented as n.s.

**Figure 6 insects-14-00551-f006:**
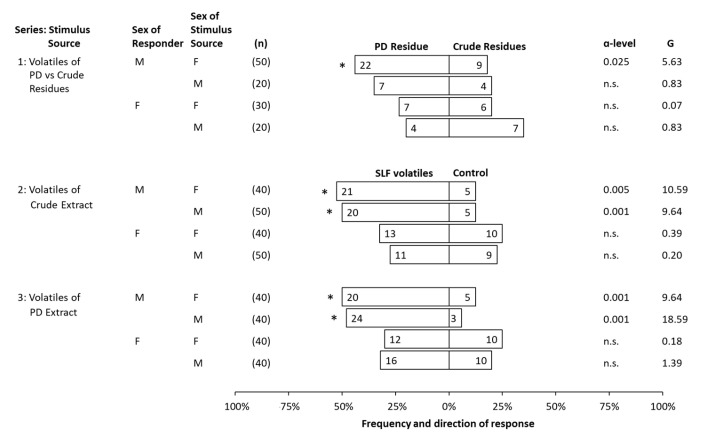
Behavioral responses made by Early adult male (M) or female (F) spotted lanternflies, *Lycorma delicatula*, when offered a choice between two stimuli. Series 1 evaluated responses in a head-to-head choice between crude and photo-degraded (PD) extract residues originating from male (M) or female (F) conspecifics (offered at 8.6 ± 0.4 insect equivalents). Series 2 and 3 evaluated adult responses to volatiles of crude or PD extracts, respectively (21.3 ± 0.9 insect equivalents), against solvent controls (dichloromethane). The total number of insects tested (n), the number of responsive insects to the respective choice (numbers in bars), ɑ-level below which *p*-values fell, and G-value are displayed for each test. The significant choices are marked with an asterisk (*), and non-significant choices are represented as (n.s.).

**Figure 7 insects-14-00551-f007:**
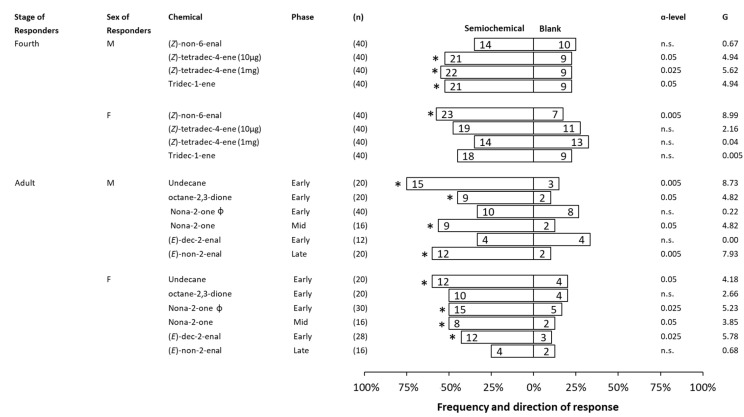
Responses in a dual-choice olfactometer made by male (M) or female (F) fourth instar or adult spotted lanternflies, *Lycorma delicatula*, when offered a choice between individual synthetic compounds (1 mg) found in body extracts and no stimulus (blank control). The total number of insects tested (n), the number of responsive insects to the respective choice, ɑ-level, and G-value are included for each test. The significant choices are marked with an asterisk (*), and non-significant choices are represented as (n.s.). Some of the behavioral data (20 out of 30 replications) for nona-2-one during Early phase were previously published [19] and denoted with Φ.

## Data Availability

The data presented in this study are openly available in Figure 1, Figure 2, Figure 3, Figure 4, Figure 5, Figure 6 and Figure 7.

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
