# Peer review of "Investigating Photo-Degradation as a Potential Pheromone Production Pathway in Spotted Lanternfly, Lycorma delicatula"

_insects, 2023, doi:10.3390/insects14060551_

Round 1
Reviewer 1 Report
1. The authors interchange between the terms 'pheromone' and 'semiochemical' throughout the manuscript. I would suggest the authors use the more general term of 'semiochemical' throughout the manuscript.
2. Line 70: Please correct this statement; sunlight is not a catalyst as it is used up during the reaction and is not regenerated at the end.
3. Line 71-77: The authors should strengthen the discussion around the chemistry involved in photodegredation; at present, not enough information is given that is of relevance to the paper, particularly the molecular mechanisms involved in radical formation and subsequent degradation pathways.
4. Line 86-98: I think this paragraph should be removed. It does not add anything to the manuscript and the important points could be summarised in one sentence later in the manuscript if needed.
5. The authors should adhere to IUPAC nomenclature when referring to chemical names. For example, under IUPAC rules, it is aluminium not aluminum.
6. Line 296-297: The authors state that they were unable to assign the position or geometric isomer of 4 compounds, however then go on to name the compounds as, for example, Z-4-tetradecene. If the authors were unable to identify the position and geometric isomer, then the compounds should not be labelled as if they were (i.e. change Z-4-tetradecene to tetradecene).
7. Figure 1: It appears that very few of the compounds analysed by GC-EAD have any EAD activity at all, with the exception of 3rd instar SLF (no stars on Figure 1). Given the apparent lack of antennal activity towards the compounds, how do the authors account for the behavioural activity?
8. Figure 1: Where is the tricosane peak that was added as an internal standard? Each of the chromatograms should be calibrated so that the size of the tricosane peak is consistent across all of the spectra to more easily allow comparison of the relative concentrations of compounds.
9. Line 349: As the authors did not determine the position or geometric isomer, they should not use them in the name of the compound.
10. Figure 3: This data is inconsistent with that presented in Figure 1. In Figure 1, the only peaks marked as antennal active are for 3rd instar larva and peak 37 in late females, while Figure 3 seems to indicate all of the compounds are antenally active for the photo-degraded samples.
11. Figure 4: What do the four bars refer too? Given the layout of Figures 5 and 6, I would assume the authors have missed off some information clarifying what each of the bars represent.
12. Line 425: How were the compounds selected to be tested on in this assay? I understand that it is not possible to test them all, but what was the rationale for selecting those compounds? Are these compounds more prevalent in the photo-treated samples? Did these compounds provide the highest EAD response?
13. Line429-438: There are some inconsistencies between the EAD data and the behavioural results. For example, the authors state that (Z)-4-tetradecene and 1-tridecene are responsive to males yet there is no EAD activity reported for those compounds in Figure 1. How do the authors explain this observation?
14. For all of the choice bioassays, the authors only present a semiochemical against a blank control. How do the authors account for the insect preference to move towards any compound compared to none (i.e. the insects are attracted to 'something' over 'nothing')? The authors should have included a control whereby an unrelated volatile compound (i.e. one not detected in the head space of either the crude or the photo-degraded) was presented against the blanks and the compounds detected in the extracts to ensure a preference for the headspace compounds over nothing.
Line 525: The authors hypothesis is that sunlight/UV light generates semiochemicals that are attractive the insects. If that is the case, then are the insects attracted to UV light, or the semiochemicals that are produced in the presence of UV light? This needs to be addressed/clarified.
The quality of English is of general good quality. Some areas of text could be improved and made more succinct though.
Author Response
Dear Reviewer,
Thank you so much for your helpful comments. We did our best to address them. Best of best.
Hajar Faal

Reviewer 2 Report
Manuscript title: Investigating photo-degradation as a potential pheromone synthesis pathway in spotted lanternfly
general comments:
The manuscript reports interesting data on a potential pheromone-production pathway of an invasive planthopper pest, Lycorma delicatula.
The results are interesting and worth publishing, however, some major issues need to be addressed before publication.
Primarily, the presentation of results should be more focused and concise and the discussion of results should also be more focused. For instance presence of a compound in a given set of samples is indicated in figures 1-3 and the compounds are also listed in the text. These results could be presented with fewer figures and should not be listed in the text as well.
State the results briefly in the Abstract and in Simple summary as well, before mentioning what they suggest.
The manuscript should be more conclusively structured, for instance GC-EAD results should be presented in the respective part of the text, not in GC-MS results.
In my opinion photo degradation of a compound is not part of its synthesis, I suggest to referring to the presumed process as pheromone production instead.
In the Introduction the authors claimed that it was necessary to develop a special technique for chemical analysis of the extracts. If this information may be of interest for other researches as well, it should be emphasized in Discussion.
Please discuss why planthoppers showed attraction towards residues as well, which presumably were missing volatile components.
specific comments:
Title: Instead of ’synthesis’ please use ’production’ instead.
Title: please indicate the species’ latin name in the title
Lines 11-12: please use the abbreviation after the English name
Lines 11-21: State the results in simple summary
Lines 22-36: State the results in the abstract
Lines 82-85: please provide information on adult categories in Materials and Methods as well
Line 163: There is no point in abbreviating ’whole body extracts’ to ’WBE’ for 1 mention in the text
Lines 242-243: Cooperband et al. is sufficient, don’t list all the authors
Line 249: please add the type of the anemometer
Lines 290-298: It is unnecessary to list information in the text which is presented in figures. Please only refer to the respective figure.
Line 335: ’Absence and presence of…’
Lines 353-354: Please indicate differences the opposite way, how much were quantities increased in photo degraded samples
Figure 4: treatments do not appear in the figure
Lines 422-438: Please explain why were the given compounds chosen for the respective life stages
Lines 495-496: Derstine et al.
Author Response

(The authors gave the same response as above.)

Reviewer 3 Report
The manuscript of Faal and collaborators titled Investigating photo-degradation as a potential pheromone synthesis pathway in spotted lanternfly reports the pheromone by L delicatula can suffer photo degradation and, the pheromone photo degraded appear to be important for short range communication, but not to long range communication. The subject is quite interesting, but there are some points that should be reviewed before publication of this study.
Introduction,
Page 2 line 59-70 the authors describe that the insects present some behaviors under sunlight, they describe that the insects during the mating prefer to stay the upper canopy, and in laboratory bioassays to orient to ultraviolet light. This behavior is quite common in all insects, bees oriented using UV light, and in general insects are attracted to light. Therefore, in my opinion this behavior did not give complete support to insects using degraded pheromones.
The authors could explore the fact that males and females do not show mating behavior during the night, and the amount of pheromone been produced during the day and night hours.
Page 2 lines 82-84 the authors defined the insects as early (adults before mating), mid (mating has begun), and late (oviposition has begun). I would like to suggest to authors to change to: early insects to virgin insects, mid insects to mature insects, and late insects change to mated insects. And this part could be described in the methodology.
Methodology
Page 4, line 168. Please provide information about the solar simulator,
Page 4 line 168 What are the ambient conditions?
The authors could describe how the volatiles were collected. What was the flow rate? A flow of 25 mL /min is too high. Was the adsorbent in a glass tube? Please provide the dimensions of the adsorbent glass tube,
Were the cuvettes containing the extracts placed into a glass chamber to collect the volatiles? Please, clarify, and give details about the chamber to collect the volatiles from the body extracts.
Page 4 line 187, what was the desired number of insects?
Line 189 there is the information that all lures were prepared with 23.3 insects equivalents, This was de desired number of insects?
Line 198. Why was used 1mg of synthetic semiochemicals?
Page 6 line 260. 20 times per session . Was this the number of replicates per treatment? The authors could clarify. What was the number of replicates for each treatment?
How were the lures connected to the Y olfactometer arms? I would like to suggest to the authors to add a figure, as supplementary material, of the olfactometer showing how the treatments were offered to insects.
Page 4 line 180. The authors could clarify that the text to parts of the body insect extracts, one called residues containing the non-volatile compounds, and the other fraction containing the volatile compounds.
GC-EAD.
How many replicates were conducted to each insect stage evaluated for GC-EAD analysis?
The authors could add a section with information of all synthetic standard used, i.e. the purity and the provider of each compound.
Results.
There is no graph or table showing the GC-EAD results. Authors could present a CG- and EAD response and a graph showing the electrophysiological responses in (–mV ± SE) of male and female and nymphs.
Figure 4 has a different layout from the other figures. All figures could be uniformized. In figure 4 add the name of the treatment like in figure 5 and 6. For all figures the authors could add in brackets the number of non-responding insects.
Discussion
Page 13 lines 500- 503. I would like to suggest removing these lines. If you evaluated the residues means that you are interested in the non-volatile compounds. In addition, you discuss along the manuscript the short and long communication.
Page 14 line 554. The authors describe that in notable that males and females’ antennae respond to the same compounds, but this response was not observed in behavioral assays. This is quite common in insect’s odor perception. The authors could explain why they report this fact as notable.
The authors also could explain the origin of these photodegraded compounds. Comparing with non-degraded compounds, for example, octane and (E)-2-hexanal were formed from what compound present in the crude extract? Most of the degraded compounds reported in this study are semiochemicals already described in other studies, and they are produced by different plants and insects playing a role in chemical communication. For example, (E)-2-hexenal is a common green leaf volatile, emitted by plants in higher level when they suffer herbivore injury, and is a defensive compound produced by several stink bugs (Weber et al., 2018). The same for (E)-2-octenal, (E)-2-decenal, these compounds are reported as defensive compounds in different species of stink bugs. In addition, the authors could comment in the discussion on the possibility of thermal degradation of some compounds, like isomerization, occurring in the inlet of the GC.
Author Response

(The authors gave the same response as above.)

Round 2
Reviewer 1 Report
I am pleased that the authors took the suggestions on board and have made the necessary adjustments to the manuscript.
Author Response
Thank you!
Reviewer 2 Report
Manuscript title: Investigating photo-degradation as a potential pheromone production pathway in spotted lanternfly, Lycorma delicatula
general comments:
The authors addressed most issues, the manuscript improved considerably. There are only some minor issues, which need to be addressed.
The identified compounds are presented in three figures and are also listed in the text, in my opinion this is unnecessary.
There is little information on EAG responses to compounds in different instars, supplementary Figure S3 does not provide information on it, please provide some more details on it.
Please explain why were different compounds chosen for behavioral tests for nymphs and adults.
specific comments:
line 46: ’were also present’
lines 406-415: please mention unidentified compounds at the end of the list
line 548: significantly more chose (?)
Author Response
Thanks for your comments.
